ecology, evolution, theoretical biology

biparental inheritance, heteroplasmy, mitonuclear, hybrid, bet-hedging, rare-allele

**Author for correspondence:**
Tom M. Allison
e-mail: tom.allison@monash.edu

# Selection for biparental inheritance of mitochondria under hybridization and mitonuclear fitness interactions

Tom M. Allison[1], Arunas L. Radzvilavicius[2] and Damian K. Dowling[1]

[1]School of Biological Sciences, Monash University, Victoria, Australia
[2]Department of Mathematics, University of Bergen, Bergen, Norway

TMA, 0000-0001-6148-1301; ALR, 0000-0002-6824-1674; DKD, 0000-0003-2209-3458

Uniparental inheritance (UPI) of mitochondria predominates over biparental inheritance (BPI) in most eukaryotes. However, examples of BPI of mitochondria, or paternal leakage, are becoming increasingly prevalent. Most reported cases of BPI occur in hybrids of distantly related sub-populations. It is thought that BPI in these cases is maladaptive; caused by a failure of female or zygotic autophagy machinery to recognize divergent male-mitochondrial DNA 'tags'. Yet recent theory has put forward examples in which BPI can evolve under adaptive selection, and empirical studies across numerous metazoan taxa have demonstrated outbreeding depression in hybrids attributable to disruption of population-specific mitochondrial and nuclear genotypes (mitonuclear mismatch). Based on these developments, we hypothesize that BPI may be favoured by selection in hybridizing populations when fitness is shaped by mitonuclear interactions. We test this idea using a deterministic, simulation-based population genetic model and demonstrate that BPI is favoured over strict UPI under moderate levels of gene flow typical of hybridizing populations. Our model suggests that BPI may be stable, rather than a transient phenomenon, in hybridizing populations.

## 1. Introduction

It is commonly held that mitochondrial DNA (mtDNA) is inherited strictly down the maternal line in most eukaryotes. From a theoretical perspective, the benefits of uniparental inheritance (UPI) of mtDNA have become increasingly clear. In modern eukaryotes, UPI is thought to suppress the spread of 'selfish' mitochondrial mutants [1,2], facilitate rapid elimination of deleterious haplotypes and fixation of beneficial haplotypes [3] and combat mutational erosion in the mitochondrial genome [4]. Despite these benefits, evidence has been growing that exceptions to UPI are widespread across the eukaryotic domain. With increasing sensitivity and deployment of deep-sequencing technologies, the last two decades have seen a marked increase in the detection of 'paternal leakage'—the inheritance of small quantities of paternal mtDNA—in animal, plant and fungal species [5–16]. Paternal leakage can generate a state of heteroplasmy, where multiple divergent mtDNA haplotypes exist within the same cell or individual [17]. While the diminutive size of sperm relative to ova strongly biases the ratio of paternal and maternal mtDNA in offspring of many plants and animals, cases of high-level intra-individual heteroplasmy (whereby two haplotypes are maintained at appreciable frequencies within an individual) have been increasingly reported [18–23].

Currently, the factors underlying the observed patterns of paternal leakage remain obscure, though most reported cases occur among hybrids of distantly related populations and subspecies [24–26]. Such cases of paternal leakage have classically been attributed to a failure of female oocytes to recognize and subsequently eliminate 'foreign' male mtDNA [26,27]. Our current mechanistic

**Figure 1.** A schematic of the population under examination in the model. Movement of diploid cells occurs between demes (with a maximum migration distance of 1 deme per generation) and mating occurs within demes. As in this schematic, figures throughout the text display the 'AA' home range on the left. Reference to the 'cline' in the text refers to the zone where both genotypes are represented (at least at a meaningful frequency), rather than the entire 50-deme stretch. (Online version in colour.)

understanding is that males attach a molecular 'tag' to sperm mitochondria which can then be recognized by female or zygotic autophagy machinery after fertilization [27]. If, however, there is little interbreeding between sub-populations, over time molecular markers will likely diverge and the molecular tag used by males in one subpopulation may become unrecognizable to the female mitochondrial destruction machinery in the other. Under this scenario, there is a breakdown in the intersexual interplay maintaining UPI, and biparental inheritance (BPI) may occur in hybrids. Despite a growing understanding of the mechanisms underpinning BPI in hybrids, currently we lack clear predictions as to whether we should expect these instances of paternal leakage to be quickly selected against and thereby transient, or beneficial and thus maintained by selection.

Recent theory has identified scenarios in which BPI may persist adaptively within species; either as a result of sexual conflict over primary control of mitochondrial inheritance (with male control favouring some paternal leakage), or because paternal leakage may mitigate the hypothesized accumulation of male-harming mtDNA variants expected under strictly maternal inheritance [4,28]. While insightful, this previous work focused on situations where there is variation in mtDNA but no variation in the mitochondrial-associated nuclear genotype (i.e. among nuclear genes with mitochondrial function) and consequently it fails to capture the dynamics in hybridizing populations where variation in both genomes may exist. Thus, we lack a clear understanding for why BPI frequently persists in sympatric populations of related taxa despite potentially considerable interpopulation gene flow [7,9,12,14,15,29–31]. While these empirical observations suggest that recurrent hybridization may itself select for BPI of mitochondria, such a contention demands rigorous assessment, particularly given the suite of theoretical benefits offered by UPI of mitochondria.

In order to capture the evolutionary dynamics of segregating variation in both nuclear and mitochondrial genomes, it is important to consider that organismal fitness is contingent upon the interaction between the two genomes. Evidence suggests that tight coordination between proteins from the two genomes is required for precise function of the electron transport system, as well as mitochondrial transcription, translation and other regulatory processes [32–35], and previous work has confirmed strong signatures of molecular coevolution between mitochondrial and nuclear genomes [32,36–40]. The consequences of disrupting coevolved mitonuclear pairs may be severe, as seen in the poor bioenergetic and phenotypic

function of experimentally engineered 'cybrid' (cyto-nuclear) hybrid individuals [41–51]. While crosses of such distantly related lineages may not be so common in wild populations, signals of mitonuclear coadaptation in wild populations exist in the form of certain patterns of mitonuclear introgression [52–54]. For example, the yellowhammer (*Emberiza citrinella*) and pine bunting (*Emberiza leucocephalos*) are hybridizing congeneric species of bird, each of which carry the same mitochondrial haplotype (probably as a result of introgression and replacement of the haplotype from one species to another) [55]. A recent study found that, in this system, nuclear genes associated with mitochondrial function were significantly overrepresented among the introgressed nuclear genes shared between the sister species, indicating a pattern of co-introgression of mitochondrial and nuclear genes from one species to the other to maintain intergenomic compatibility [55].

In the present paper, we test the hypothesis that more efficient matching between mitochondrial and nuclear gene pairs may be facilitated by allowing BPI of mitochondria. Accordingly, we explore whether BPI may be selected for under certain rates of hybridization, thus helping to explain previous empirical reports of persistent paternal leakage in sympatric hybridizing populations. Thus, we define a hypothetical 'mitochondrial inheritance mode' allele that encodes BPI when distantly related individuals mate and UPI when closely related individuals mate. We term this allele 'mate-specific BPI', and herein provide a mathematical model competing strict UPI against mate-specific BPI, to assess whether such an allele would be favoured or eliminated in hybridizing populations, under an assumption of a mitonuclear interactive effect on fitness.

## 2. The model

In order to assess the adaptive value of a mate-specific BPI trait in a hybrid zone, we observe a metapopulation of single-celled, diploid organisms comprised two distinct lineages ('populations' hereafter). The populations are arranged along a one-dimensional cline consisting of 50 'demes': sub-populations wherein individuals mate and between which individuals migrate (figure 1). We track a single mitochondrial locus with two alleles—0 and 1—and an interacting nuclear (denoted N-mt) locus—also with two alleles $A$ and $a$—such that each population is characterized by a particular coevolved combination ($A/0$ and $a/1$, respectively).

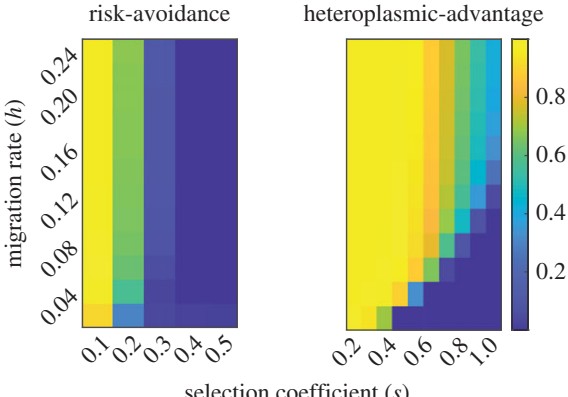

**Figure 2.** Mean frequency of the *u* allele throughout the entire cline at equilibrium or after 75 000 generations; whichever came first. The *u* allele is injected into a metapopulation of hybridizing cells at a frequency of 0.01 after migration/selection equilibrium has been reached along the cline. It is then able to evolve freely. For the parameter space explored, *u* invades more readily under the heteroplasmic-advantage model than the risk-avoidance model. Yellow represents a high equilibrium frequency of *u* and blue represents little or no invasion, as denoted by the scale on the right-hand side of the figure. In each case, the conditions where *u* reaches a high equilibrium frequency correspond to areas of low selection and higher migration, where gene flow is high. Parameter values: diploid mitochondrial number $M = 100$, number of demes $C = 50$. (Online version in colour.)

We also track a second nuclear locus that controls the inheritance mode of mitochondria. One allele (*U*) codes for strict UPI, whereas the other (*u*) codes for mate-specific BPI. This allele is completely linked with a sex-determination locus, such that it is only expressed in females at the gamete stage. We consider linkage of sex-determination and mitochondrial inheritance mode loci representative of many eukaryotic systems [56–60], however, there may be some systems (particularly multicellular organisms with heterogametic males) without such strict linkage that allow for males to carry and spread an inheritance mode allele without expressing it. Modelling both systems shows that strict linkage offers the more conservative estimate for the spread of the proposed inheritance mode allele (figure 2; electronic supplementary material, figure S2), and thus strict linkage formed the focus of the present analysis. We believe that deeper analysis of the dynamics underlying spread of the *u* allele in the 'recombination' case may provide fruitful avenues for future research.

In the model, female gametes with the *U* allele always reject their partner's mitochondria and the full number of mitochondria in the offspring is restored by replication of existing mitochondria. Female gametes with the *u* allele will accept their partner's mitochondria *if and only if* their N-mt alleles *do not* match (i.e. *A* meets *a*). Otherwise, zygote formation will proceed with UPI as normal. This is consistent with our current understanding of mitochondrial recognition systems, which depend on expression of nuclear genes for the molecular tag that is attached to sperm [26,27].

Each cell contains *M* mitochondria in the diploid (zygote/adult) stage. Our analyses focus on the situation when $M = 100$, since varying *M* did not have a large impact on the spread of the *u* allele (electronic supplementary material, figure S3). Fitness is determined by the proportion of each mitochondrial haplotype given a particular nuclear background. It is assessed in the diploid stage and thus its description requires three distinct fitness curves (one for

each of the possible N-mt genotypes—*AA*, *Aa* and *aa*). We model two different sets of fitness curves in order to test two distinct hypotheses for the beneficial effects of a putative mate-specific BPI allele (electronic supplementary material, figure S1). Under both model variations, homozygote fitness follows a concave curve, meaning that fitness declines more rapidly with each successive addition of a mismatched mitochondrion. This kind of fitness curve is thought to best explain the observation of 'threshold' effects, where heteroplasmic individuals display mild symptoms up to a critical frequency of 'mutant' mitochondria, after which disease may be severe [61].

The first hypothesis—which we term the 'risk-avoidance' hypothesis—is based solely on the idea that BPI of mitochondria reduces the risk of producing descendants with a complete mitonuclear mismatch. This idea is best tested by attributing a flat cost to all hybrids (i.e. there is no fitness benefit for being in a state of heteroplasmy *per se*). Any fitness advantage gained through BPI and subsequent heteroplasmy in this scenario is necessarily manifested solely in increased fitness of homozygous descendants.

The second hypothesis—which we term the 'heteroplasmic-advantage' hypothesis—proposes that the BPI of mitochondria brought about by reproduction between individuals of divergent lineages may confer direct fitness gains to hybrids because it ensures that the haploid contribution of N-mt genes transmitted by each parent is paired to at least some matching mitochondria. We see this hypothesis belonging to a broader class of evolutionary models wherein an allele may increase in frequency in a hybrid zone if it improves hybrid fitness (or rather minimizes outbreeding depression) and carries a sufficiently small cost to homozygotes of the parent populations [62]. To assess this idea, we explicitly incorporate some additional benefit to a hybrid possessing both types of mitochondria. Hybrids still suffer a fitness cost relative to homozygotes, but we assume a relative fitness benefit for heterozygotes with intermediate amounts of each mitochondrial haplotype (see electronic supplementary material, figure S1 for fitness curves). In each variation, strength of selection may be modified using a selection coefficient *s*, which varies between 0 and 1, with larger values representing a more drastic mitonuclear mismatch.

A complete model cycle is characterized by selection (on diploid cells), migration of surviving cells between demes (at rate *h*), meiosis (with recombination between the N-mt locus and inheritance mode locus) and finally random mating within demes. For precise description of these processes, see Methods in electronic supplementary material.

We begin the simulation with a metapopulation where *AA* individuals populate the first 25 demes and *aa* individuals populate the remaining demes and allow it to reach migration/selection balance. We proceed by injecting the *u* allele at a frequency of 0.01 across the cline (into each deme). Although it may be more biologically plausible to assume that paternal leakage is the default mode and show that UPI does *not* invade such a population, we thought it a stronger display of the adaptive benefits of mate-specific BPI to show this trait invading a population. The ultimate goal remains to show the conditions under which BPI of mitochondria may be beneficial. The simulation was stopped when *u* reached an equilibrium frequency (defined as a frequency change of less than $10^{-10}$ between generations) or after 75 000 generations, whichever came first. We tracked invasion of the mate-specific

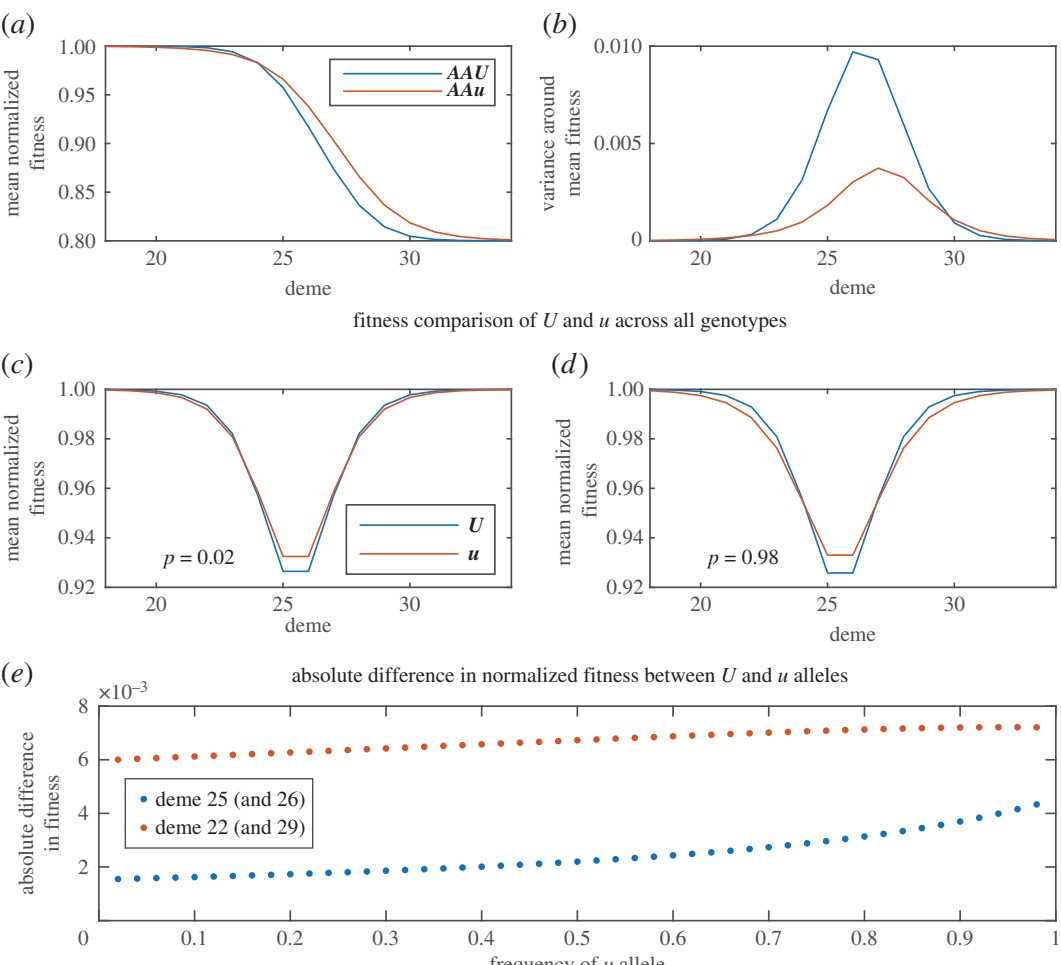

**Figure 3.** Comparison of the fitness of $U$ and $u$ individuals aggregated over specific N-mt genotypes and over the entire population. (*a*) In regions of the cline where the *A* allele is more common, $u$ carries a cost to *AA* individuals and in regions of the cline where the *A* allele is less common, $u$ provides an advantage. (*b*) $u$ individuals have lower variance in fitness in the cline centre since they have a much lower chance of producing either a mismatched 'low fitness' genotype or a fully matched 'high fitness' genotype. (*c,d*) Mean normalized fitness of the $U$ and $u$ alleles averaged across all genotypes under low frequency of $u$ (*c*) and high frequency of $u$ (*d*). $u$ offers a net advantage in the middle two demes (25 and 26) and a disadvantage in the demes either side of the centre. This trend is exaggerated as the frequency of $u$ increases (from *c,d*). While the difference in the cline centre increases roughly linearly with increasing frequency of $u$ (red line, (*e*)), the difference outside the cline centre increases quadratically (blue line, (*e*)). Parameter values: $M = 100$, $C = 50$, migration rate $h = 0.1$, selection strength $s = 0.2$ (moderate gene flow). Fitness curves: 'risk-avoidance'. (Online version in colour.)

BPI allele $u$ under a wide range of migration ($h$) and selection ($s$) values, though we limit migration to $0 < h < 0.25$, such that there are no cases where more than half of the individuals leave the deme in which they were formed.

# 3. Results

We found that the frequency of the $u$ allele ($p$) always converges on a single stable equilibrium $0 \leq p \leq 1$ under the parameter space explored (figure 2). In the hetero-plasmic-advantage model, $u$ invades under the majority of migration/selection combinations, whereas in the risk-avoid-ance model $u$ invades in a smaller region of parameter space. In each case, the parameter space where the allele is more likely to invade corresponds to lower selection strength and higher migration, where gene flow is higher. Our results show that $u$ is always beneficial to hosts that also possess the uncommon N-mt allele for a given region of the cline (figure 3*a*) and that higher gene flow allows this benefit to be realized in more cells (due to increased proportions of the uncommon allele), thus explaining the link between gene flow and

invasion of the mate-specific BPI allele. Under the heteroplas-mic-advantage model, increasing hybrid fitness can allow for an additional increase in frequency of the uncommon allele in the opposing subpopulation's range, in turn furthering the range of demes where the benefits of $u$ can be realized. For this reason, along with the simple additive benefits of increased hybrid fitness when in a state of heteroplasmy, $u$ invades over a larger portion of parameter space under the heteroplasmic-advantage model.

## (a) Risk-avoidance model

To investigate the reasons underlying spread or elimination of $u$ under the risk-avoidance model, we examine some basic population statistics while keeping migration rate constant and varying selection strength; and vice versa by varying migration rate while keeping the selection strength constant. We also examine the effects of varying $p$ in the metapopulation by keeping $u$ at a fixed frequency during simulations.

Both migration rate and selection strength strongly influ-ence the distribution of N-mt alleles $A$ and *a* along the cline (electronic supplementary material, figure S4). Increasing

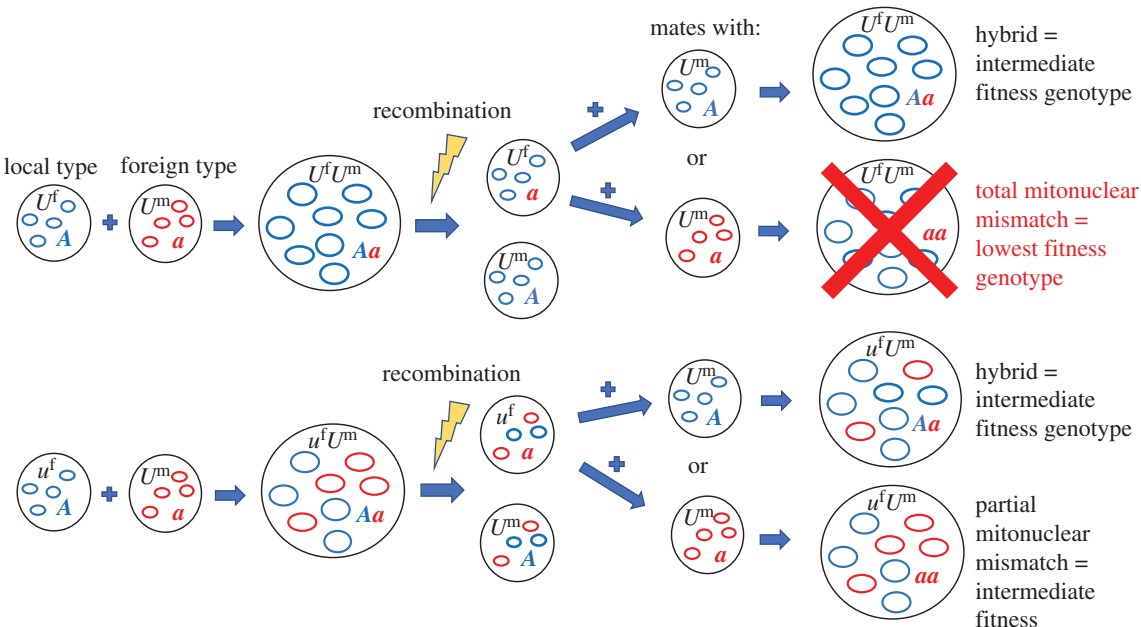

**Figure 4.** A schematic of how, through recombination, an *AU* gamete in its 'home range' risks eventually producing a totally mismatched *aaU* zygote when paired with a rare *a* allele. This is contrasted with the lower variance pathway of an *Au* gamete following the same path but producing descendants all of intermediate fitness. This scenario is played out in the '*A* side' of the cline, where *A* and 1 haplotypes (represented by blue circles) are the most common and *a* and 0 haplotypes (red circles) are relatively rare. (Online version in colour.)

selection strength has a similar effect to decreasing migration rate: each reduces gene flow. The effect is a narrower cline and more unequal gene frequencies in the two adjacent demes comprising the cline centre (demes 25 and 26) (electronic supplementary material, figure S4A,C). The two may act in concert to produce particularly high or low gene flow (electronic supplementary material, figure S4E). Examining the frequency of the 'majority' N-mt allele in the cline centre over multiple fixed frequencies of *u*, we see that while migration and selection strongly determine the distribution of genotypes, the frequency of *u* plays a comparatively small role (electronic supplementary material, figure S4B,D,F).

Varying *u*, however, strongly affects the distribution of mitochondrial alleles (electronic supplementary material, figure S5). In the cline centre, heteroplasmy increases with increasing *u*, even approaching the theoretical limit where all cells exhibit a 50 : 50 mix of each haplotype. Since heteroplasmic cells cannot achieve either maximum or minimum possible fitness, increasing the frequency of *u* reduces variance in fitness. Thus, *fitness* of each N-mt genotype across the cline is affected by both migration/selection ratio and frequency of *u*. By comparing cells of the same N-mt genotype and different mitochondrial inheritance modes (for example, *AAU* versus *AAu*),[1] we see that there are some regions in the cline where *u* is beneficial to its host and others where it is detrimental (figure 3*a*).

Homozygotes with strict UPI perform equally well or better than mate-specific BPI homozygotes when on 'their side' of the cline (where the N-mt allele they possess is the most common). This is true for all migration/selection combinations explored. When a particular N-mt allele is overwhelmingly more common, any mate-specific BPI gametes encountering an (albeit rare) opposing N-mt allele will result in heteroplasmy and—given the high probability that this allele finds itself in a homozygote in the next generation—sentence any descendants to a reduction in maximum fitness. Note that for this to happen, there must be at least some probability of a gamete encountering

an opposing N-mt allele. This explains why the fitness differences are most notable a few demes on either side of the cline centre and taper off to almost zero towards either end of the cline (corresponding to regions of moderate and very low likelihood of encountering a mismatched N-mt allele, respectively).

Conversely, having the mate-specific BPI allele provides an advantage in homozygotes when the opposing N-mt allele is the dominant type (*AAu* individuals perform better than *AAU* individuals in a predominantly *a* environment). We can imagine *a* allele gametes with strict UPI in a predominantly *a* environment (genotype *aU*): upon encountering an *A* gamete and forming a zygote, subsequent recombination means that half of these encounters will produce a gamete with *A* at the N-mt locus, *U* at the inheritance mode locus and all type 1 mitochondria (a completely mitonuclear-mismatched gamete). Any subsequent encounter between the now mismatched *A*/1 gamete and another gamete will at best produce a hybrid with intermediate fitness and at worst (upon encountering another rare *A* allele) a mismatched homoplasmic homozygote—the lowest fitness genotype. The benefit of *u*, therefore, is in avoiding this situation. There is very little chance of producing a mismatched homoplasmic zygote since any pairing between an '*au*' gamete and an *A* gamete will result in heteroplasmy, as demonstrated in the idealized schematic in figure 4. Thus, while *u* precludes maximum fitness, it also prevents formation of the lowest fitness genotypes. In other words, populations with the *U* genotype will experience greater variance in fitness when compared to populations of the same N-mt genotype but bearing the *u* allele (figure 3*a*).

The result of tensions between benefits of *u* when paired with the rarer N-mt allele and the costs of *u* when paired with the common N-mt allele is a net benefit for *u* in the cline centre (demes 25 and 26) and a net cost in the demes on either side (figure 3*c*,*d*). Note that this is only true for cases where *u* invades, as there is no net benefit of *u* in the cline centre under low gene flow due to a lower rate of interactions

between $A$ and $a$ alleles. These trends are merely exaggerated by increasing the frequency of $u$. But while the benefit of $u$ in the cline centre increases in a roughly linear fashion, the cost of $u$ on either side increases quadratically with increasing frequency of the $u$ allele (figure 3$e$). Thus, an invasion may begin with some demes overproducing $u$ and the allele spreading, but as the magnitude of benefits and costs change under the growing frequency of $u$, elimination of $u$ in some regions may catch up to production in others. A stable intermediate frequency is reached when net production in some regions equals the net elimination in others. Due to persistent net production in the cline centre and net elimination elsewhere, we see a hump shape in the clinal distribution of the $u$ allele during invasion and at equilibrium (electronic supplementary material, figure S6).

## (b) Heteroplasmic-advantage model

The reasons for spread of $u$ under the heteroplasmic-advantage model are broadly similar, such that many of the results already discussed also apply to this model. There are, however, two key differences. Firstly, since heterozygotes can achieve higher fitness with heteroplasmy, N-mt genotype frequencies may be more strongly altered by increasing frequency of $u$ (electronic supplementary material, figure S7). Essentially, under higher frequencies of $u$, increased hybrid fitness means that the N-mt alleles and their associated mitochondria are able to persist further into the other population's range (the cline is widened). Particularly under moderate gene flow, increasing the frequency of $u$ in the metapopulation increases the cline width and produces a more even ratio of each homozygote in the cline centre. This broadens the regions where 'risk-avoidance' advantages may be realized.

Secondly, the heteroplasmic advantage itself contributes greatly towards increasing fitness of those cells possessing the $u$ allele. This acts in an additive fashion upon the already established benefits of risk-avoidance. This additive benefit is enough to frequently overturn a deme from net elimination to net production of the $u$ allele. As a consequence of these two differences acting in conjunction, the migration/selection conditions under which $u$ invades are much greater (figure 2).

## 4. Discussion

We sought to provide a mathematical test of the hypothesis that BPI of mitochondria confers an adaptive advantage during episodes of hybridization between populations. The hypothesis is based on the assumption that when individuals from divergent populations hybridize, interactions between mitochondrial and nuclear gene products may disrupt electron transport system function and mitochondrial regulation systems, causing a loss of phenotypic function [32,38,63,64]. Our results show that when divergent populations hybridize under higher rates of gene flow, mate-specific BPI may benefit populations by avoiding the production of completely mitonuclear-mismatched organisms. The implication is that when populations hybridize under certain levels of gene flow, we may expect BPI of mitochondria to be stable and adaptive, as opposed to maladaptive and rapidly selected against.

We show that the net benefits of a mate-specific BPI allele are always highest in the cline centre, where the likelihood of encountering a gamete from a foreign population is greatest. This implies that the risk of producing a mitonuclear-

mismatched cell is underpinned by uncertainty in the origin of potential mating partners. We can imagine that when prezygotic barriers to reproduction are yet to develop and mate-selection is random, there is a high risk of choosing a partner that leads to a poor mitonuclear match in the contact zone. Though the uncertainty in mate-selection is not affected by possessing the mate-specific BPI allele, the state of heteroplasmy that is associated with this allele in the hybrid zone means that individuals with this genotype have a lower risk of producing descendants with a total mitonuclear mismatch. We show that under weak to moderate selection, a genotype that sacrifices the highest fitness phenotype in return for avoiding the lowest fitness phenotype will enjoy higher mean fitness and hold an evolutionary advantage.

We show a putative mate-specific BPI allele invading a population of strict UPI cells, however, it is important to note that under hybridization through secondary contact, mate-specific BPI may already be the default state [26]. Biologically, we refer to cases where molecular tags, which are normally attached to male mtDNA to signal destruction, have diverged in the intervening period of isolation and BPI proceeds due to a failure of female recognition systems. Our results suggest that if mate-specific BPI is already present in hybridizing populations due to diverged molecular recognition systems—and providing that this divergence is also sufficient to disrupt interactions between mitochondrial and nuclear gene products—then any allele restoring the recognition systems required for strict UPI may facilitate worse mitonuclear matches and be swiftly eliminated. In the light of this, a testable prediction coming from our model is that the divergence of mitochondrial recognition and destruction mechanisms should diverge between populations at a faster rate than expected by drift alone.

Mathematically, however, the distinction between a mate-specific BPI allele invading a population of strict UPI individuals and a strict UPI allele invading a population of mate-specific BPI individuals is redundant, as our results show a single stable equilibrium frequency of $u$ throughout the cline (in other words, the starting frequency of mate-specific BPI alleles does not affect the equilibrium frequency). This means that our results offer new predictions for the occurrence of BPI in clines where hybridization is the result of environmental gradients rather than secondary contact. Our current understanding of how mitochondrial molecular recognition systems fail does not address situations where mitochondrial divergence has occurred despite persistent gene flow between hybridizing populations; as may be the case under spatially driven local adaptation. Under this type of hybridization, we may not *a priori* expect that male-mitochondrial recognition systems would diverge at all. If, however, an environmental gradient supports a mitochondrial polymorphism and there is subsequent compensatory nuclear evolution, our model predicts that we should in fact expect to see evolution away from strict UPI towards mate-specific BPI (provided there is enough migration between the two sub-populations). This is a particularly interesting prediction, as it suggests that paternal leakage could be driven by the requirement for sustaining mitochondrial function, rather than by allopatric divergence alone. We suggest that further empirical investigations into rates of heteroplasmy in hybrid zones along environmental gradients known to shape mitochondrial evolution, such as temperature and altitude [65–67], will be insightful.

We also demonstrate that if being in a state of heteroplasmy reduces outbreeding depression, BPI will be beneficial over a wide range of migration and selection conditions. The requisite assumption may at first seem extreme since some empirical findings and theoretical models suggest that heteroplasmy might reduce fitness [3,68]. We note, however, that empirical studies have only been carried out on nuclear-homozygous organisms and that there is a lack of empirical evidence comparing fitness consequences of homoplasmy and heteroplasmy in hybrids. We suggest here that if both nuclear alleles are expressed then each mitochondrion could potentially receive the correctly matched nuclear-encoded mitochondrial products. Indeed, it is thought that mitochondria-specific molecular recognition sites aid in targeting nuclear-encoded products to the required organelle [33,69,70], raising the intriguing prospect that a haplotype could 'request' products specifically from its matched nuclear allele.

As an alternative explanation, in the case of metazoans with multiple tissues, possessing two distinct mitochondrial haplotypes in conjunction with their respective coevolved nuclear alleles in a single individual may provide more variation upon which tissue-specific selection of mitochondria can act. The idea of differential selection on components of mitochondrial function across tissues remains an open question [35], though there are some examples [71–74]. Given that mitochondrial proteomes differ greatly between tissues [75], differential selection for mitochondrial function leaves open the possibility of tissue-specific segregation of mitochondrial haplotypes. In this case, having two sets of mitochondrial haplotypes would always be *at least* as good as one set (if one set performed best in all tissues) and perhaps better if different sets functioned best in different tissues. The net result, in the latter case, would be a higher mean fitness across all tissues and probably a higher individual fitness. We suggest that further empirical studies into tissue-specific selection of mitochondria based on function will probably provide greater insight into this hypothesis.

While the heteroplasmic-advantage model remains more speculative than the risk-avoidance model, it still provides an interesting example of how a trait reducing outbreeding depression can be selected for in a hybrid zone, even if it carries costs to homozygotes of the parent populations. Previous modelling work has demonstrated that such a trait can be selected for

within the central hybrid zone without increasing in frequency at all in the rest of the cline [62]. Our study offers new insights into this under-explored side of hybridization [76]. While previous work has modelled fixed costs for homozygotes of parent populations, we show that allowing variation in the fitness of homozygotes throughout the cline (in our case depending on the load of each mitochondrial haplotype and in turn the frequency of the invading $u$ allele) can in some cases allow spread of the invading allele throughout the entire population. In other words, the mate-specific BPI trait we present here is a trait that reduces outbreeding depression, carries costs for homozygotes of the parent populations, but is not necessarily a 'rare-allele' as classically presented [62,77,78]. Thus, our results provide support for the adaptive value of alleles reducing outbreeding depression without providing support for the rare-allele phenomenon—suggesting that these 'rare' alleles may be even more common than empirical findings currently suggest.

Data accessibility. The MATLAB code used for simulations is available as electronic supplementary material, along with a data file showing simulation results.

Authors' contributions. T.M.A.: conceptualization, formal analysis, investigation, methodology, visualization, writing—original draft; A.L.R.: Investigation, methodology, supervision, writing—review and editing; D.K.D.: conceptualization, investigation, project administration, supervision, validation, writing—review and editing. All authors gave final approval for publication and agreed to be held accountable for the work performed therein.

Competing interests. We declare we have no competing interests.

Funding. This research was supported through the Australian Research Council's Discovery Projects funding scheme (project no. DP210102931). T.M.A. was supported by a scholarship from the Australian Government Research Training Program.

Acknowledgements. We are grateful to Tim Connallon, Sean Layh, Craig White and Luke McKenzie-McHarg for thoughtful discussions and mathematical insights.

## Endnote

[1]For statistical representation of the metapopulation in Results we only consider the inheritance mode locus ($U/u$) when it is expressed, which only occurs in females. Although both alleles are incorporated into the model for males, for this analysis we average the fitness over both $U$ and $u$ males. This leaves us with six functionally different genotypes to assess, rather than the 16 of a classic two-locus model.

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
