## [Peer Review File · Proceedings of the Royal Society B: Biological Sciences]

Review History

RSPB-2021-1600.R0 (Original submission)

Review form: Reviewer 1

Recommendation

Accept with minor revision (please list in comments)

Scientific importance: Is the manuscript an original and important contribution to its field?

Excellent

General interest: Is the paper of sufficient general interest?

Good

Quality of the paper: Is the overall quality of the paper suitable?

Excellent

Is the length of the paper justified?

Yes

Should the paper be seen by a specialist statistical reviewer?

No

Do you have any concerns about statistical analyses in this paper? If so, please specify them explicitly in your report.

No

It is a condition of publication that authors make their supporting data, code and materials available - either as supplementary material or hosted in an external repository. Please rate, if applicable, the supporting data on the following criteria.

Is it accessible?

Yes

Is it clear?

Yes

Is it adequate?

Yes

Do you have any ethical concerns with this paper?

No

Comments to the Author

Mito-nuclear epistasis, paternal leakage, and positive or negative selection of those combinations are incredibly important, complex, and overlooked issues in evolutionary genetics. Although there is significant literature on each of these concepts alone, little has been done to address them in a single model, which the authors have clearly done here and hopefully this is the beginning of a significant expansion on these topics.

Toward that end, I would encourage the authors to spend a paragraph or so detailing 1 or 2 real-world examples of sub-species introgression of mito-nuclear pairs that support mito-nuclear adaptation to emphasize the importance of this and not just mito-nuclear mismatch as the important process typically associated with the interaction of the two genomes. For example, the ABC brown bears in Alaska carry polar bear mitochondrial DNA from an ancient introgression and also carry a larger proportion of polar bear X-chromosome than expected. This suggests mito-nuclear co-adaptation if only male brown bears migrated from the mainland to the islands to mate with female polar bears, and if both male and female brown bears migrated to the islands, it suggests selection for the entire mito-nuclear complex.

Review form: Reviewer 2

Recommendation

Accept with minor revision (please list in comments)

Scientific importance: Is the manuscript an original and important contribution to its field?

Good

General interest: Is the paper of sufficient general interest?

Good

Quality of the paper: Is the overall quality of the paper suitable?

Excellent

Is the length of the paper justified?

Yes

Should the paper be seen by a specialist statistical reviewer?

No

Do you have any concerns about statistical analyses in this paper? If so, please specify them explicitly in your report.

No

It is a condition of publication that authors make their supporting data, code and materials available - either as supplementary material or hosted in an external repository. Please rate, if applicable, the supporting data on the following criteria.

Is it accessible?

Yes

Is it clear?

Yes

Is it adequate?

Yes

Do you have any ethical concerns with this paper?

No

Comments to the Author

This article by Allison et al explores the conditions under which biparental inheritance of mitochondria (BPI) evolves as a response to selection pressures due to hybridization. Most organisms have in fact uniparental inheritance of mitochondria (UPI) but as the authors point out more instances of paternal leakage or some degree of BPI are being reported. In addition, the role of mitonuclear coadaptation whereby the nuclear and mitochondrial genomes must match one another for optimal fitness have been the focus of several experimental and theoretical studies. Here, the authors ask whether mate specific BPI whereby gametes from different populations meet (while UPI is maintained within populations) would evolve. The tension here lies in the fact that UPI is known to increase variance and facilitate selection for the optimal mitochondrial-nuclear types, whereas BPI ensures that some agreement between mitochondrial and nuclear genomes persist and/or that heteroplasmic mitochondria genomes are avoided. The authors analyze a mathematical model that explores the invasion of a mate specific BPI allele as a response to these selection pressures. They find that mate specific BPI evolves under the above selection pressures and explore and discuss the conditions under which BPI between populations is stable.

The article is very well written and I enjoyed going through it. I think the model provides an original and interesting insight to our current state of understanding about the way in which mitochondrial genomes evolve and are inherited. I have a few, minor, questions and suggestions that may help clarify a few points that I list below.

1. As far as I understand no asexual growth is considered in the model. Asexual growth is likely to increase variation in fitness in the population and so mitigate any fitness costs due to hybridization. What do the authors think the impact of including asexual growth would be on their conclusions?
2. The number of mitochondria in the model is kept fixed. It would be useful to see how the size of the mitochondria pool impact on the authors' analysis.
3. I would suggest adding a diagram at the beginning of the article that outlines the life cycle and spatial structure of the model, where authors also graphically introduce what they mean by cline and deme.
4. Page 17, lines 329-333. This is an interesting point. I wonder if the authors' results also

suggest that the divergence of recognition mechanisms for mitochondria destruction would diverge between populations at a rate faster than that expected by drift? If so this would provide an additional testable prediction from the model.

Decision letter (RSPB-2021-1600.R0)

29-Sep-2021

Dear Mr Allison:

Your manuscript has now been peer reviewed and the reviews have been assessed by an Associate Editor. The reviewers' comments (not including confidential comments to the Editor) and the comments from the Associate Editor are included at the end of this email for your reference. As you will see, the reviewers and the Editors have raised some concerns with your manuscript and we would like to invite you to revise your manuscript to address them.

Research ethics:

Use of animals and field studies:

It is a condition of publication that you make available the data and research materials supporting the results in the article. Please see our Data Sharing Policies (<https://royalsociety.org/journals/authors/author-guidelines/#data>). Datasets should be deposited in an appropriate publicly available repository and details of the associated accession number, link or DOI to the datasets must be included in the Data Accessibility section of the article (<https://royalsociety.org/journals/ethics-policies/data-sharing-mining/>). Reference(s) to datasets should also be included in the reference list of the article with DOIs (where available).

Please submit a copy of your revised paper within three weeks. If we do not hear from you within this time your manuscript will be rejected. If you are unable to meet this deadline please let us know as soon as possible, as we may be able to grant a short extension.

Best wishes,
Dr Daniel Costa
<mailto:proceedingsb@royalsociety.org>

Associate Editor
Comments to Author:

In this paper, the authors develop a mathematical model to explore the idea that rather than a byproduct of species divergence, biparental inheritance of mitochondria seen in hybrids can actually be under positive natural selection. Both reviewers found the paper interesting, important and well written, and I completely agree with this assessment.

I only have one comment that should be addressed in addition to the reviewers' comments. I was initially a little confused by the sentence in l.127-128, probably because I'm not used to systems where sex is determined at the haploid stage. Eventually it made sense but it made me wonder to

what extent the results would be translatable to multicellular organisms that are either hermaphrodites or have separate sexes. If, in such systems, u is only expressed in the eggs but is not sex-linked (i.e., can also end up in sperm without doing anything there), would the results still hold? I think this would be an important question to discuss given that many readers will be more interested in animals or plants and might assume the results are readily applicable to those systems as well.

Reviewer(s)' Comments to Author:

Referee: 1

Comments to the Author(s)

Mito-nuclear epistasis, paternal leakage, and positive or negative selection of those combinations are incredibly important, complex, and overlooked issues in evolutionary genetics. Although there is significant literature on each of these concepts alone, little has been done to address them in a single model, which the authors have clearly done here and hopefully this is the beginning of a significant expansion on these topics.

Toward that end, I would encourage the authors to spend a paragraph or so detailing 1 or 2 real-world examples of sub-species introgression of mito-nuclear pairs that support mito-nuclear adaptation to emphasize the importance of this and not just mito-nuclear mismatch as the important process typically associated with the interaction of the two genomes. For example, the ABC brown bears in Alaska carry polar bear mitochondrial DNA from an ancient introgression and also carry a larger proportion of polar bear X-chromosome than expected. This suggests mito-nuclear co-adaptation if only male brown bears migrated from the mainland to the islands to mate with female polar bears, and if both male and female brown bears migrated to the islands, it suggests selection for the entire mito-nuclear complex.

Referee: 2

Comments to the Author(s)

This article by Allison et al explores the conditions under which biparental inheritance of mitochondria (BPI) evolves as a response to selection pressures due to hybridization. Most organisms have in fact uniparental inheritance of mitochondria (UPI) but as the authors point out more instances of paternal leakage or some degree of BPI are being reported. In addition, the role of mitonuclear coadaptation whereby the nuclear and mitochondrial genomes must match one another for optimal fitness have been the focus of several experimental and theoretical studies. Here, the authors ask whether mate specific BPI whereby gametes from different populations meet (while UPI is maintained within populations) would evolve. The tension here lies in the fact that UPI is known to increase variance and facilitate selection for the optimal mitochondrial-nuclear types, whereas BPI ensures that some agreement between mitochondrial and nuclear genomes persist and/or that heteroplasmic mitochondria genomes are avoided. The authors analyze a mathematical model that explores the invasion of a mate specific BPI allele as a response to these selection pressures. They find that mate specific BPI evolves under the above selection pressures and explore and discuss the conditions under which BPI between populations is stable.

The article is very well written and I enjoyed going through it. I think the model provides an original and interesting insight to our current state of understanding about the way in which mitochondrial genomes evolve and are inherited. I have a few, minor, questions and suggestions that may help clarify a few points that I list below.

1. As far as I understand no asexual growth is considered in the model. Asexual growth is likely to increase variation in fitness in the population and so mitigate any fitness costs due to hybridization. What do the authors think the impact of including asexual growth would be on their conclusions?
2. The number of mitochondria in the model is kept fixed. It would be useful to see how the size of the mitochondria pool impact on the authors' analysis.

3. I would suggest adding a diagram at the beginning of the article that outlines the life cycle and spatial structure of the model, where authors also graphically introduce what they mean by cline and deme.
4. Page 17, lines 329-333. This is an interesting point. I wonder if the authors' results also suggest that the divergence of recognition mechanisms for mitochondria destruction would diverge between populations at a rate faster than that expected by drift? If so this would provide an additional testable prediction from the model.

Author's Response to Decision Letter for (RSPB-2021-1600.R0)

See Appendix A.

Decision letter (RSPB-2021-1600.R1)

08-Nov-2021

Dear Mr Allison

I am pleased to inform you that your Review manuscript RSPB-2021-1600.R1 entitled "Selection for biparental inheritance of mitochondria under hybridisation and mitonuclear fitness interactions" has been accepted for publication in Proceedings B.

The referee(s) do not recommend any further changes. Therefore, please proof-read your manuscript carefully and upload your final files for publication. Because the schedule for publication is very tight, it is a condition of publication that you submit the revised version of your manuscript within 7 days. If you do not think you will be able to meet this date please let me know immediately.

To upload your manuscript, log into <http://mc.manuscriptcentral.com/prsb> and enter your Author Centre, where you will find your manuscript title listed under "Manuscripts with Decisions." Under "Actions," click on "Create a Revision." Your manuscript number has been appended to denote a revision.

You will be unable to make your revisions on the originally submitted version of the manuscript. Instead, upload a new version through your Author Centre.

- 1) A text file of the manuscript (doc, txt, rtf or tex), including the references, tables (including captions) and figure captions. Please remove any tracked changes from the text before submission. PDF files are not an accepted format for the "Main Document".
- 2) A separate electronic file of each figure (tiff, EPS or print-quality PDF preferred). The format should be produced directly from original creation package, or original software format. Please note that PowerPoint files are not accepted.
- 3) Electronic supplementary material: this should be contained in a separate file from the main text and the file name should contain the author's name and journal name, e.g. `authorname_procb_ESM_figures.pdf`

All supplementary materials accompanying an accepted article will be treated as in their final form. They will be published alongside the paper on the journal website and posted on the online figshare repository. Files on figshare will be made available approximately one week before the accompanying article so that the supplementary material can be attributed a unique DOI. Please see: <https://royalsociety.org/journals/authors/author-guidelines/>

4) Data-Sharing and data citation

It is a condition of publication that data supporting your paper are made available. Data should be made available either in the electronic supplementary material or through an appropriate repository. Details of how to access data should be included in your paper. Please see <https://royalsociety.org/journals/ethics-policies/data-sharing-mining/> for more details.

If you wish to submit your data to Dryad (<http://datadryad.org/>) and have not already done so you can submit your data via this link <http://datadryad.org/submit?journalID=RSPB&manu=RSPB-2021-1600.R1> which will take you to your unique entry in the Dryad repository.

Once again, thank you for submitting your manuscript to Proceedings B and I look forward to receiving your final version. If you have any questions at all, please do not hesitate to get in touch.

Sincerely,
Dr Daniel Costa
Editor, Proceedings B
<mailto:proceedingsb@royalsociety.org>

Associate Editor Board Member

Comments to Author:

The authors have done an excellent job addressing the (very few) issues raised by the reviewers' and myself.

However, it appears as if something has gone wrong with the figures in this revision. The labels have shifted and look overly large in the new Fig. 1 as well as Fig.4 (which looked fine in the old version), and the figure captions are missing. (The version without track changes doesn't have any figures at all, and no figure captions either.)

Also, I think the sentence in l.101-104 needs a reference, and the following sentences might be better linked to this one. (It sounds at first as if this about a different system, not the same bird system.)

Decision letter (RSPB-2021-1600.R2)

11-Nov-2021

Dear Mr Allison

I am pleased to inform you that your manuscript entitled "Selection for biparental inheritance of mitochondria under hybridisation and mitonuclear fitness interactions" has been accepted for publication in Proceedings B.

Data Accessibility section

Open Access

Paper charges

Sincerely,

Proceedings B

Appendix A

Dear Editors,

We thank you and the referees for the constructive feedback on our manuscript - we are delighted that you and the reviewers enjoyed reading our work. The questions and comments that have been raised are very insightful, and we have conducted further analyses of our model in response, and incorporated revisions that address these comments directly into the manuscript. Below, we've listed each editor / referee comment, along with our responses and details of modifications to the revised manuscript. We believe the revised manuscript is much improved as a result of the review process. We hope you find our responses helpful and satisfying; please let us know should you have any further queries.

Best wishes,

Tom Allison

Associate Editor

“If [...] u is only expressed in the eggs but is not sex-linked (i.e., can also end up in sperm without doing anything there), would the results still hold?”

Early modelling efforts on this paper incorporated precisely the proposed system – where the sex-determination locus and inheritance-mode locus were able to recombine freely. This system in fact allows even greater spread of the u allele (albeit tending to fix under different regions of parameter space). Given that a system where sex-determination and mitochondrial inheritance mode loci are closely linked is likely representative of many eukaryotes, we decided that the more conservative system should form the focus of our analysis - though we do believe that future modelling efforts to understand the dynamics of spread under recombination would be valuable. We have included a figure in the supplementary material showing the equilibrium frequencies of u under the case of recombination between sex-determination and inheritance-mode loci. Since this question may be raised by other readers, we have also added a brief summary explaining this choice to readers and referring them to supplementary material (Figure S2) to see the results of the alternative (recombination) case (pasted below).

“We consider linkage of sex-determination and mitochondrial inheritance-mode loci representative of many eukaryotic systems (Fedler *et al.* 2009; Shakya & Idnurm 2014; Speijer *et al.* 2015; Coelho *et al.* 2018; Sun *et al.* 2020), however there may be some systems (particularly multicellular organisms with heterogametic males) without such strict linkage that allow for males to carry and spread an inheritance-mode allele without expressing it. Modelling both systems shows that strict linkage offers the more conservative estimate for the spread of

the proposed inheritance mode allele (Figure 2 and Figure S2 in Electronic Supplementary Material (ESM)), and thus strict linkage formed the focus of the present analysis. We believe that deeper analysis of the dynamics underlying spread of the u allele in the ‘recombination’ case may provide fruitful avenues for future research. “

Reviewer 1:

“Mito-nuclear epistasis, paternal leakage, and positive or negative selection of those combinations are incredibly important, complex, and overlooked issues in evolutionary genetics. Although there is significant literature on each of these concepts alone, little has been done to address them in a single model, which the authors have clearly done here and hopefully this is the beginning of a significant expansion on these topics.”

We thank the referee for their review, and are delighted that they appreciate the impact of our work, and that it is clear.

“...Toward that end, I would encourage the authors to spend a paragraph or so detailing 1 or 2 real-world examples of sub-species introgression of mito-nuclear pairs that support mito-nuclear adaptation to emphasize the importance of this and not just mito-nuclear mismatch as the important process typically associated with the interaction of the two genomes.”

We recognise that in our presentation of the interaction between mitochondrial and nuclear genomes, we did focus heavily on the potential for purifying selection as the dominant evolutionary force. In an attempt to instead emphasize the importance of mito-nuclear adaptation, we have altered paragraph 4 (lines 93 – 108) to include a recent example of a very interesting case of mito-nuclear introgression in birds. Our minor modifications in lines 110-111 were also made with the aim of rectifying the heavy focus on ‘mismatch’.

Reviewer 2:

“The article is very well written and I enjoyed going through it. I think the model provides an original and interesting insight to our current state of understanding about the way in which mitochondrial genomes evolve and are inherited”

We thank the referee for their review, and are honoured to read that our work is recognised for its originality in this field.

“1. As far as I understand no asexual growth is considered in the model. Asexual growth is likely to increase variation in fitness in the population and so mitigate any fitness costs due to hybridization. What do the authors think the impact of including asexual growth would be on their conclusions?”

It is very difficult to propose an answer without running the model (which would require extensive work). On one hand, under asexual reproduction, the strict uniparental inheritors do not run the risk of a total mito-nuclear mismatch each generation and would be free to reproduce at maximum fitness (unless, of course, we include mutations, setting the stage for muller's ratchet to crank). However, we built a case throughout the paper that the risk of a total mito-nuclear mismatch is highest when there is strong mixing of cells that differ greatly in their mitochondrial content – which would be exacerbated under bouts of sexual contact between spells of asexual growth. Our speculation would be that the benefits of mate-specific BPI would hold, except for under long periods of asexual growth (ie. 20+ generations of asexual reproduction between rounds of sexual reproduction), at which point the benefits of mate-specific BPI may well be reduced. We suggest, however, that future efforts modelling this system would be better spent trying to understand the impact of asexual growth in the context of multicellularity (ie. the germ/soma distinction and the resultant higher between-cell variance in the germline), which we contend would not impact our conclusions. To see why, look to column 5 of the schematic in figure 4 – even if all gametes coming from the zygote in column 3 were to become homoplasmic (as may happen in the germline of an initially heteroplasmic metazoan), only gametes with the strict uniparental inheritance mode run the risk of producing a fully mismatched zygote (which could not be ameliorated by further asexual divisions). This, however, would require far more extensive modelling, however, and is beyond the scope of this project.

“2. The number of mitochondria in the model is kept fixed. It would be useful to see how the size of the mitochondria pool impact on the authors’ analysis.”

We excluded mitochondrial number from the model initially since the analysis was already quite demanding. We chose the number (50/100 haploid/diploid) to maximise quantity, whilst keeping computation times reasonable (simulation run-time increases non-linearly with ‘cell-size’). It would take longer than the time granted for revisions to run simulations for larger cell sizes. Instead, to get an indication of how mitochondrial number might influence the analysis, we ran the simulations with fewer mitochondria (20/40 haploid/diploid), giving us the ability to at least tentatively extrapolate findings. We found that results were very similar between these simulations and those published in the main paper. Given that other readers may raise the same question, we have referred readers to Supplementary Material (Figure S3) for the results when $M = 40$ (see changes in lines 173-175).

“3. I would suggest adding a diagram at the beginning of the article that outlines the life cycle and spatial structure of the model, where authors also graphically introduce what they mean by cline and deme”

We have added these clarifications Please see the updated Figure 1 and legend.

“4. Page 17, lines 329-333. This is an interesting point. I wonder if the authors’ results also suggest that the divergence of recognition mechanisms for mitochondria destruction would

diverge between populations at a rate faster than that expected by drift? If so this would provide an additional testable prediction from the model.”

This is an insightful point. In fact, we consider it an even more powerful insight coming from the model than our own proposed conclusion. As such, we have replaced our conclusion with the suggestion from the reviewer (see lines 390 - 393).